# Mesh Denoising Using Filtering Coefficients Jointly Aware of Noise and Geometry

## ABSTRACT

Mesh denoising is a fundamental task in geometry processing, and recent studies have demonstrated the remarkable superiority of deep learning-based methods in this field. However, existing works commonly rely on neural networks without explicit designs for noise and geometry which are actually fundamental factors in mesh denoising. In this paper, by jointly considering noise intensity and geometric characteristics, a novel Filtering Coefficient Learner (FCL for short) for mesh denoising is developed, which delicately generates coefficients to filter face normals. Specifically, FCL produces filtering coefficients consisting of a noise-aware component and a geometry-aware component. The first component is inversely proportional to the noise intensity of each face, resulting in smaller coefficients for faces with stronger noise. For the effective assessment of the noise intensity, a noise intensity estimation module is designed, which predicts the angle between paired noisy-clean normals based on a mean filtering angle. The second component is derived based on two types of geometric features, namely the category feature and face-wise features. The category feature provides a global description of the input patch, while the face-wise features complement the perception of local textures. Extensive experiments have validated the superior performance of FCL over state-of-the-art works in both noise removal and feature preservation.

## CCS CONCEPTS

• **Computing methodologies** → **Mesh models**; *Shape representations*; *Shape analysis*.

## KEYWORDS

Mesh denoising, filtering, noise intensity, geometric characteristics.

## 1 INTRODUCTION

In recent years, the acquisition of meshes from real-world objects has become increasingly accessible thanks to advancements of 3D scanning equipment and reconstruction algorithms [7, 13, 16, 32]. However, even with advanced techniques, meshes obtained from real-world objects are inevitably contaminated by noise, which can cause inefficiencies or even failures in downstream geometric tasks. Consequently, mesh denoising has emerged as a fundamental research topic in geometry processing [4, 18, 21, 33, 38].

**Unpublished working draft. Not for distribution.**

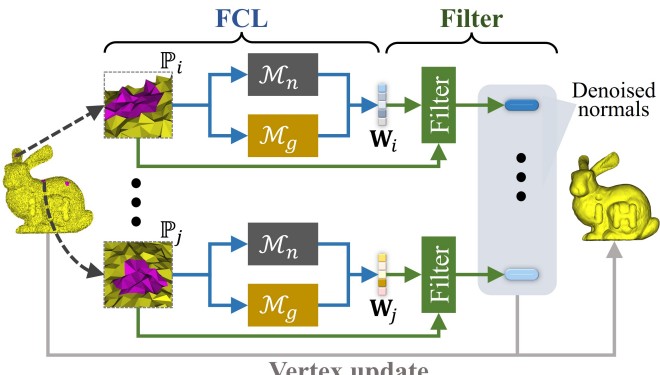

**Figure 1: The workflow of the proposed method.** $\mathbb{P}$ **stands for a local patch. W denotes the learned filtering coefficients.**

Mesh denoising aims to smooth a noisy surface while simultaneously preserving the underlying geometric features, which is an ill-posed inverse problem [23, 36]. Overcoming this, conventional methods [6, 12, 30, 34, 37] usually rely on some assumptions about underlying features and noise patterns. But these assumptions are difficult to generalize across various meshes and noise [26], limiting the performance of conventional methods. Recently, deep-learning-based methods [11, 14, 23, 27, 29, 36] have been proposed to predict noise-free face normals for mesh denoising. These methods typically take a local mesh patch as input to predict the noise-free normal. Since meshes are irregular, general convolutional networks are not directly applicable to meshes. Previous works have elegantly addressed this issue through ingenious representation of meshes. For example, Zhao *et al.* [36] employ 3D convolutions to regress noise-free normals from the voxel-based representation of local mesh patches. Li *et al.* [11] apply a network similar to Point-Net++ [20] to regress the denoised normal from a patch of face normals. Shen *et al.* [23] infer denoised normals through graph convolutions which accept a graph representation on the dual space of mesh faces as input. However, although deep learning-based methods have achieved remarkable performance without relying on specific assumptions, none of these networks incorporate explicit designs for noise and geometry which are actually fundamental factors in mesh denoising

In this paper, we propose a novel Filtering Coefficient Learner (FCL for short), which delicately produces filtering coefficients aware of noise and geometry for mesh denoising. Inspired by the success of combining deep learning with filtering in denoising tasks [17, 28], FCL is designed to output filtering coefficients instead of denoised normals. The complete mesh denoising procedure contains three steps: coefficient learning, filtering, and vertex updating, as depicted in Fig. 1. The coefficients consist of a noise-aware component and a geometry-aware component. The noise-aware

component is inversely proportional to the noise intensity of each face. The stronger the noise on a face, the smaller the corresponding coefficient for its normal. For the effective assessment of the noise intensity, a noise intensity estimation module ($\mathcal{M}_n$) is designed, which predicts the angle between paired noisy-clean normals based on a mean filtering angle. The geometry-aware component is derived by a geometry describing module ($\mathcal{M}_g$) based on two types of geometric features, namely the category feature and face-wise features. The category feature is extracted through a classifier that categorizes faces into four groups, providing a global description of the input patch. On the other hand, the face-wise features are automatically captured by graph convolutions, complementing the perception of local textures.

The main contributions of this paper can be summarized as follows:

- By jointly considering noise intensity and geometric characteristics, a novel Filtering Coefficient Learner (FCL for short) is proposed, which delicately generates coefficients to filter face normals.
- A noise intensity estimation module is designed to derive the noise-aware component of filtering coefficients through predicting the angle between paired noisy-clean normals. The generated coefficients are inversely proportional to the noise intensity of each face.
- A geometry describing module is developed to capture comprehensive geometric features for producing the geometry-aware component of filtering coefficients. The geometric features contain the category feature which offers a global description of the input patch, and face-wise features that complement the perception of local textures.

## 2 RELATED WORKS

Mesh denoising has been a fundamental research topic in geometry processing for many years, leading to the development of various denoising methods. In this section, we provide a comprehensive analysis and review of filter-based and deep-learning-based mesh denoising methods that are most relevant to the proposed approach.

## 2.1 Filter-based Methods

Filter-based methods are widely used in feature-preserving mesh denoising due to their effectiveness and simplicity. The pioneering works [1, 2, 5, 6, 8, 22, 31] in this area are heavily inspired by 2D image denoising techniques [3, 25] and applied directly to vertices. For example, Fleishman *et al.* [6] and Jones *et al.* [8] employ bilateral filters to adjust vertex positions directly. However, the vertex-filtering mode is found to be limited by the fact that face normals are better at revealing local geometry than vertices [9, 37]. In light of this, a series of works [10, 34, 35, 37] that first denoise face normals and then update vertex positions achieve better denoising performance. Zhang *et al.* [34] propose a joint bilateral filter that takes the averaged normal of a most consistent local patch as the guidance information. Li *et al.* [10] apply the corner-aware neighborhood to derive the guidance normals, which do better in adapting to complex features than [34]. Zhao *et al.* [35] propose to compute a guidance normal field with the graph-cut algorithm, and then use the guidance field to filter normals.

In summary, filter-based methods have been the dominant approach in feature-preserving mesh denoising methods. However, the common limitation is that the coefficients are derived based on assumptions which are difficult to generalize across various meshes and noise. In contrast, the proposed FCL learns coefficients dynamically according to noise intensity and geometric characteristics. It does not rely on any specific assumptions, realizing improved performance and more robust denoising results.

## 2.2 Deep-learning-based Methods

3D meshes are irregular, which makes general convolutional neural networks not directly applicable [11, 14, 23]. Therefore, designing appropriate networks to elegantly learn mesh features has always been the focus of deep-learning-based mesh denoising techniques. Pioneering works adopt hand-crafted features or voxel representations for feature learning. Wang *et al.* [27] introduce a filtered face normal descriptor (FND) based on the bilateral filter with multiple kernels. FND is then fed into simple multi-layer perceptrons for noise-free normal regression. Li *et al.* [14] propose to represent mesh patches using non-local patch-group normal matrices (NPNMs). They first learn low-rank NPNMs, and then feed the finetuned NPNMs into a 2D convolutional network to predict noise-free normals. Zhao *et al.* [36] develop a voxel-based representation for local mesh patch, enabling the use of 3D convolutions to regress noise-free normals. These methods with hand-crafted features or voxel representations inevitably suffer from insufficient or redundant information. To address this drawback, subsequent works prefer end-to-end networks. Li *et al.* [11] apply a network similar to PointNet++ [20] to estimate the denoised normal with a patch of face normals as input. This is the first end-to-end network for mesh denoising. Shen *et al.* [23] represent mesh patches in a graph form, which naturally captures the geometry features. The patch graphs are fed into a graph convolution network to infer denoised normals. This scheme is not only end-to-end, but also preserves complete geometric information.

Previous works have elegantly addressed the issue caused by the irregularity of meshes, achieving superior performance over conventional methods. However, all these methods apply networks without explicit designs for noise and geometry which are actually fundamental factors in mesh denoising. In contrast, FCL is designed by jointly considering noise intensity and geometric characteristics.

## 3 METHODOLOGY

The proposed FCL is utilized for mesh denoising following a three-step paradigm. For each face in a noisy mesh, FCL takes its local patch as input to learn filtering coefficients first. The learned coefficients are then used to derive the denoised normal for each face. Once all the denoised normals have been obtained, the vertex positions are accordingly updated using a well-studied scheme [23, 24, 37].

As shown in Fig. 2, FCL is composed of a noise intensity estimation module ($\mathcal{M}_n$) and a geometry describing module ($\mathcal{M}_g$). $\mathcal{M}_n$ generates the noise-aware component of filtering coefficients, while $\mathcal{M}_g$ produces the geometry-aware component. This section begins with a problem statement regarding normal denoising and

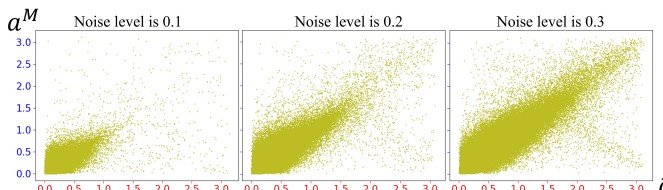

**Figure 2: The structure of FCL. The arrows serve as visual indicators for different processes. The green arrow (↑) depicts the filtering process. The blue arrow (↑) represents the generation process of the noise-aware component. The brown arrow (↑) indicates the process of capturing the category feature. The red arrow (↑) signifies the production process of the geometry-aware component.**

then provides detailed explanations of $\mathcal{M}_n$ and $\mathcal{M}_g$, respectively. Finally, the vertex updating scheme is briefly introduced.

## 3.1 Normal Denoising Problem Statement

A mesh containing $N_v$ vertices and $N_f$ faces is expressed as $\mathcal{M} = \{\mathbb{V}, \mathbb{F}\}$, where $\mathbb{V} = \{\mathbf{v}_i\}_1^{N_v}$ is the set of vertices while $\mathbb{F} = \{f_i\}_1^{N_f}$ is the set of faces. For each face $f_i \in \mathbb{F}$, its normal is denoted as $\mathbf{n}_i$ and its centroid is $\mathbf{c}_i$. FCL takes the $r$-ring patch of $f_i$ as input to learn the filtering coefficients:

$$\mathbf{W}_i = \mathrm{FCL}(\mathbb{P}_i). \tag{1}$$

Here, $\mathbb{P}_i$ is the $r$-ring patch of $f_i$, and $\mathbf{W}_i = [w_1, w_2, ..., w_{|\mathbb{P}_i|}]^T$ is the filtering coefficient vector. The denoised normal is generated by filtering the normals of faces in $\mathbb{P}_i$:

$$\mathbf{n}_i' = \frac{1}{|\mathbb{P}_i|} \sum_{f_j \in \mathbb{P}_i} w_j * \mathbf{n}_j. \tag{2}$$

For a face $f_i$, its patch $\mathbb{P}_i$ is initialized to $\{f_i\}$, and is generated by iteratively adding all the faces that share at least one vertex with the faces in $\mathbb{P}_i$ for $r$ times [36]. For clarity, $r$ is set to 3 in this paper. To remove unnecessary degrees of freedom from the input patch, we translate $\mathbb{P}_i$ to the origin, scale it into a unit bounding box, and rotate it to the direction where the mean normal of $\mathbb{P}_i$ is $[0, 0, 1]$.

## 3.2 Noise Intensity Estimation Module

The noise intensity estimation module ($\mathcal{M}_n$) produces the noise-aware component (denoted as $\mathbf{W}^n$), which is related to the noise intensity of each face. This subsection explains the ground truth, input, and structure of $\mathcal{M}_n$ successively

**Ground Truth**. The noise intensity of a face can be effectively assessed by the angle between its normal and the corresponding ground truth normal:

$$\hat{a}_i = \mathrm{acos}(\mathbf{n}_i \cdot \hat{\mathbf{n}}_i), \tag{3}$$

where $\hat{a}_i$ is the angle and $\hat{\mathbf{n}}_i$ is the ground truth normal. Larger angle suggests stronger noise. The ground truth counterpart of $\mathbf{W}^n$ is calculated as:

$$\hat{\mathbf{W}}^n = [\hat{w}_1^n, \hat{w}_2^n, ..., \hat{w}_{|\mathbb{P}|}^n]^T$$
$$= [1 - \frac{\hat{a}_1}{\pi}, 1 - \frac{\hat{a}_2}{\pi}, ..., 1 - \frac{\hat{a}_{|\mathbb{P}|}}{\pi}]^T. \tag{4}$$

In this way, every $\hat{w}^n$ is between 0 and 1. The stronger the noise of $f_i$, the smaller the $\hat{w}_i^n$. $\mathcal{M}_n$ is trained to estimate $\hat{\mathbf{W}}^n$.

**Input**. The input of $\mathcal{M}_n$ includes a novel mean filtering angle ($a^M$) proposed in this paper. Fig. 3 shows the relation between $\hat{a}$ and $a^M$. With the x-axis representing $\hat{a}$ and the y-axis being $a^M$, we can clearly see that $a^M$ and $\hat{a}$ are approximately linear with each other. In light of this, we take advantage of $a^M$ to estimate $\hat{w}^n$ that is calculated based on $\hat{a}$. As a result, the normals ($\mathbf{N}_i$) and centroids ($\mathbf{C}_i$) of faces in a patch are concatenated with the mean filtering angles ($\mathbf{A}_i^M$) as the input of $\mathcal{M}_n$.

For each face $f_i$, its $a_i^M$ is the angle between $\mathbf{n}_i$ and a filtered normal. The filtered normals are face normals of the mesh (denoted

**Figure 3: The relationship between $a^M$ and $\hat{a}$.**

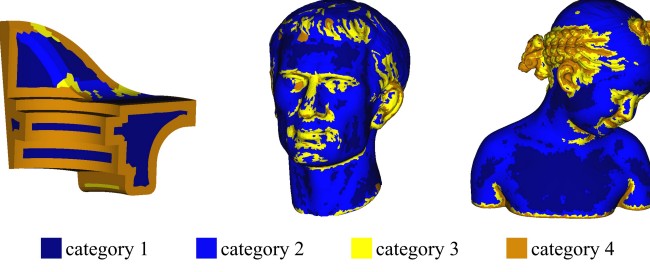

| category 1 | category 2 | category 3 | category 4 |

**Figure 4: Example meshes with categorized faces.**

as $\mathcal{M}^M$) obtained by performing a simple mean filter on $\mathcal{M}$. $\mathcal{M}^M$ is generated through iterative normal filtering and vertex updating. For each face $f_i$, the normal filtering is conducted as:

$$\mathbf{n}_i^{k+1} = \frac{1}{|\mathbb{P}_i^f|} \sum_{f_j \in \mathbb{P}_i^f} \mathbf{n}_j^k. \tag{5}$$

Here, $\mathbb{P}_i^f$ contains all faces that share an edge with $f_i$. $k$ denotes the $k$-th iteration. Subsequently, vertex positions are updated according to the obtained normals. In our method, $\mathcal{M}^M$ is obtained through 20 iterations. Finally, the mean filtering angle is calculated as follows:

$$a_i^M = \mathrm{acos}(\mathbf{n}_i \cdot \mathbf{n}_i^M). \tag{6}$$

**Structure of $\mathcal{M}_n$.** As show in Fig. 2, $\mathcal{M}_n$ follows a simple structure. First, three static graph convolutional layers ([128, 128, 256]) are employed to capture face-wise features from the input. Then, a max-pooling and an average-pooling are used to capture global features. Finally, a multi-layer perceptron ([512, 512, 128, 1]) regresses $\mathbf{W}^n$ from the multi-scale features composed of face-wise features and global features. Like in [11, 19, 20], the multi-layer perceptron acts separately on each face with shared parameters to solve the problem of face disorder.

### 3.3 Geometry Describing Module

The geometry describing module ($\mathcal{M}_g$) produces the geometry-aware component of filtering coefficients based on a category feature and face-wise features. In this subsection, we first elaborate on the extraction of the category feature, and then explain the structure of $\mathcal{M}_g$, which involves the capturing of face-wise features.

**Category feature.** During the training of a mesh denoising network, a common challenge arises due to the imbalanced distribution of data among different categories of patches, such as flat, edge, and corner patches. To address this issue, previous works [23, 36] typically divide all faces into four categories and randomly select an equal number of samples from each category for training. It is intuitive to consider that the filtering coefficients for patches of different categories should be different as well. Therefore, we employ a classifier to extract the category feature and integrate it into the learning of the geometry-aware component. For each face $f_i$, its category label is generated based on the maximum angle difference within its 2-ring patch, following a similar approach as in [36]. Denoting the maximum angle difference as $A$, all the faces in $\mathbb{F}$ are divided into four categories:

category 1: $\ 0° < A \leq \ 20°$, smooth region

category 2: $20° < A \leq \ 50°$, curved region
category 3: $50° < A \leq \ 80°$, small edge region
category 4: $80° < A \leq 180°$, large edge region

Three example meshes with classified faces are shown in Fig. 4. It is worth mentioning that we conducted experiments with more than four categories, but unfortunately, we did not observe any additional benefits from increasing the number of categories. The corresponding experiments are available in Subsection 5.4.

The classifier is indicated by brown arrows in Fig. 2. It utilizes three graph convolutional layers ([128, 128, 256]) along with symmetric pooling operations to capture global features, following a similar structure as $\mathcal{M}_n$. Subsequently, three fully connected layers ([256, 256, 4]) are employed to regress the category probability. The first two FC layers in the classifier capture the category feature, while the last layer outputs the category probability only used for training.

**Structure of $\mathcal{M}_g$.** As show in the bottom part of Fig. 2, the normals ($\mathbf{N}_i$) and centroids ($\mathbf{C}_i$) of faces in a patch are fed into $\mathcal{M}_g$. The face-wise features captured by GCN are concatenated with the category feature to obtain comprehensive description of the input patch. Then, the comprehensive features are fused through a multi-layer perceptron ([512, 512]). The first row of the fused feature map corresponds to the face in processing, which is called as the central feature. All features in the patch are multiplied by the central feature to obtain the similarity between each face and the central face. Finally, three fully connected layers ([128, 128, 128]) take the similarity vector as input and output $\mathbf{W}_i^g$.

### 3.4 Vertex Updating

The vertex updating scheme employed in our method follows the approach outlined in [23, 37]. To compute the updated position $\mathbf{v}_i'$, we consider the neighboring faces of vertex $\mathbf{v}_i$, denoted by the set $\mathbb{P}_i^\mathbf{v}$. This set includes all faces that contain $\mathbf{v}_i$ as one vertex. Mathematically, it is expressed as:

$$\mathbf{v}_i' = \mathbf{v}_i + \frac{1}{|\mathbb{P}_i^\mathbf{v}|} \sum_{f_j \in \mathbb{P}_i^\mathbf{v}} \mathbf{n}_j'(\mathbf{n}_j' \cdot (\mathbf{c}_j - \mathbf{v}_i)). \tag{7}$$

$\mathbf{n}_j'$ is the denoised normal of $f_j$, and $\mathbf{c}_j$ is the centroid of $f_j$. The equation computes the updated vertex position $\mathbf{v}_i'$ by summing up the contribution from each neighboring face. The contribution is determined by the dot product between the denoised normal $\mathbf{n}_j'$ and the vector $(\mathbf{c}_j - \mathbf{v}_i)$, which measures the displacement from the vertex $\mathbf{v}_i$ to the centroid $\mathbf{c}_j$ of the face $f_j$. The resulting sum is then averaged by the number of neighboring faces $|\mathbb{P}_i^\mathbf{v}|$.

## 4 TRAINING

The training of FCL is guided by three loss functions in three stages. In this section, we introduce the loss functions first, and then explain the training scheme.

### 4.1 Loss Function

For each face $f_i$, the input patch is denoted as $\mathbb{P}_i$. The first loss function guides the parameter optimization of $\mathcal{M}_n$. Since the production of $\mathbf{W}^n$ is a regression problem, we choose the mean square

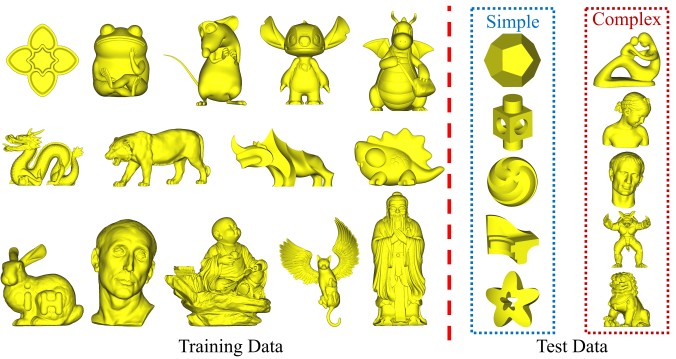

Training Data        Test Data

Figure 5: The triangle meshes in SynData.

error as the loss function:

$$L_{\mathcal{M}_n} = \frac{1}{|\mathbb{P}_i|} \sum_{f_j \in \mathbb{P}_i} (w_j^n - \hat{w}_j^n)^2. \tag{8}$$

Here, $\mathbb{P}_i$ represents the input patch, $w_j^n$ denotes the predicted filtering coefficient for face $f_j$, and $\hat{w}_j^n$ is the corresponding ground truth value. The second loss function is a cross entropy loss function used for training the classifier. This loss function ensures that the predicted category labels are consistent with the ground truth labels. Mathematically, it is expressed as:

$$L_{cls} = -\frac{1}{|\mathbb{P}_i|} \sum_{f_j \in \mathbb{P}_i} \sum_{k=1}^{4} p_{j,k} \log(q_{j,k}). \tag{9}$$

Here, $p_{j,k}$ is the predicted probability that $f_j$ belongs to category $k$, while $q_{j,k}$ is the corresponding label. The last loss function measures the cosine distance between the denoised normal $\mathbf{n}_i'$ and the ground truth normal $\hat{\mathbf{n}}_i$. The purpose of this loss function is to ensure that the denoised normals accurately represent the true geometry. It is defined as:

$$L_{normal} = 1 - \cos(\mathbf{n}_i', \hat{\mathbf{n}}_i) \tag{10}$$

### 4.2 Training Scheme

During the training process, $\mathcal{M}_n$ is trained in the first stage using the loss function $L_{\mathcal{M}n}$, followed by training the category classifier in the second stage using $L_{cls}$. In the last stage, the entire FCL is trained using the weighted sum of all three loss functions:

$$L = (1 - \lambda) * L_{normal} + \lambda * (L_{\mathcal{M}_n} + L_{cls}). \tag{11}$$

Here, the parameter $\lambda$ controls the relative importance of the normal loss ($L_{normal}$) compared to the combination of the $\mathcal{M}n$ loss ($L_{\mathcal{M}n}$) and the category classifier loss ($L_{cls}$). The setting of $\lambda$ is experimentally conducted and the experimental results are put in Subsection 5.4. For clarity, there is no parameter freezing operation during the entire training process. This allows for the joint optimization of the network and ensures that all components effectively contribute to the denoising performance.

## 5 EXPERIMENTS

In this section, we present the experimental setup, comparison studies, ablation studies, and investigations on hyper-parameters.

Table 1: The experimental results on SynData.

| Methods | Simple meshes | | Complex meshes | |
|---|---|---|---|---|
| | $E_a$ | $E_v(\times 10^{-3})$ | $E_a$ | $E_v(\times 10^{-4})$ |
| Noisy | 24.51 | 3.62 | 24.74 | 12.77 |
| BMF [6] | 7.15 | 2.91 | 7.23 | 9.95 |
| BNF [37] | 4.94 | 2.32 | 6.96 | 8.69 |
| GNF [34] | 4.76 | 2.34 | 6.85 | 9.14 |
| TGV [15] | **3.84** | 3.01 | 5.43 | 10.95 |
| GCN [23] | 4.86 | 2.41 | 5.25 | 8.21 |
| Ours | 4.57 | **2.29** | **4.84** | **8.11** |

### 5.1 Experimental Setup

**Dataset**. FCL is evaluated on both synthetic and real-scanned datasets. The synthetic dataset (denoted as SynData) is composed of 3D triangle meshes collected from [23], [15], and an online 3D model repository (3dmag.org). SynData consists of 14 training meshes and 10 test meshes (5 simple geometric meshes and 5 complex object meshes), as shown in Fig. 5. Noisy meshes for training are generated by adding Gaussian noise (the standard deviations are 0.1, 0.2, and 0.3 of the mesh average edge length) and impulsive noise (the numbers of impulsive vertices are 10%, 20%, and 30% of the mesh vertex numbers). The test set only covers Gaussian noise.

The real-scanned datasets encompass the Kinect series datasets [27], as well as meshes obtained from the internet. The Kinect series datasets (Kv1Data, Kv2Data, and K-FData) are obtained by scanning six objects (*big girl*, *cone*, *girl*, *boy*, *David*, and *pyramid*) using Microsoft Kinect v1 and v2. The meshes from the internet include *angel*, *eagle*, *gargoyle 1*, *gargoyle 2*, *Lucy*, and *rabbit*.

**Implementation details.** In the training of FCL, the truncated normal distribution is used to initialize the weights. The optimizer is Adam with the default parameter settings ($\beta_1 = 0.9$, $\beta_2 = 0.9$, $\epsilon = 10^{-8}$) in PyTorch. We set the batch size to 512. The first two training stage are conducted for 500 epochs, while the last stage is for 1000 epochs The learning rate starts at 0.01 and decays by half after the 300th, 700th, 800th, 900th epochs. The training process is executed on a computer equipped with an AMD Ryzen 9 5900HX CPU and an NVIDIA GeForce RTX 3080 Laptop GPU. On the SynData, 10000 faces are randomly selected from each noisy mesh to participate in training in every epoch. On the three Kinect datasets, 1000 faces are randomly selected from each noisy point cloud in every epoch.

**Error metric**. Two commonly adopted metrics are used in our experiments. $E_a$ measures the average normal angular difference between a denoised mesh and the ground truth noise-free mesh:

$$E_a = \frac{1}{N_f} \sum_{f_i \in \mathbb{F}} \mathrm{acos}(\mathbf{n}_i' \cdot \hat{\mathbf{n}}_i). \tag{12}$$

Here, $\mathbb{F}$ is the set of faces, while $N_f$ is the number of faces in $\mathbb{F}$. $\mathbf{n}_i'$ and $\hat{\mathbf{n}}_i$ are the denoised normal and ground truth normal of $f_i$, respectively. $E_v$ is the normalized average Hausdorff distance from the denoised mesh to the corresponding ground-truth mesh [27]:

$$E_v = \frac{1}{N_v} \sum_{\mathbf{v}_i' \in \mathbb{V}} \min_{\hat{\mathbf{v}}_i \in \hat{\mathbb{V}}} ||\mathbf{v}_i' - \hat{\mathbf{v}}_i||. \tag{13}$$

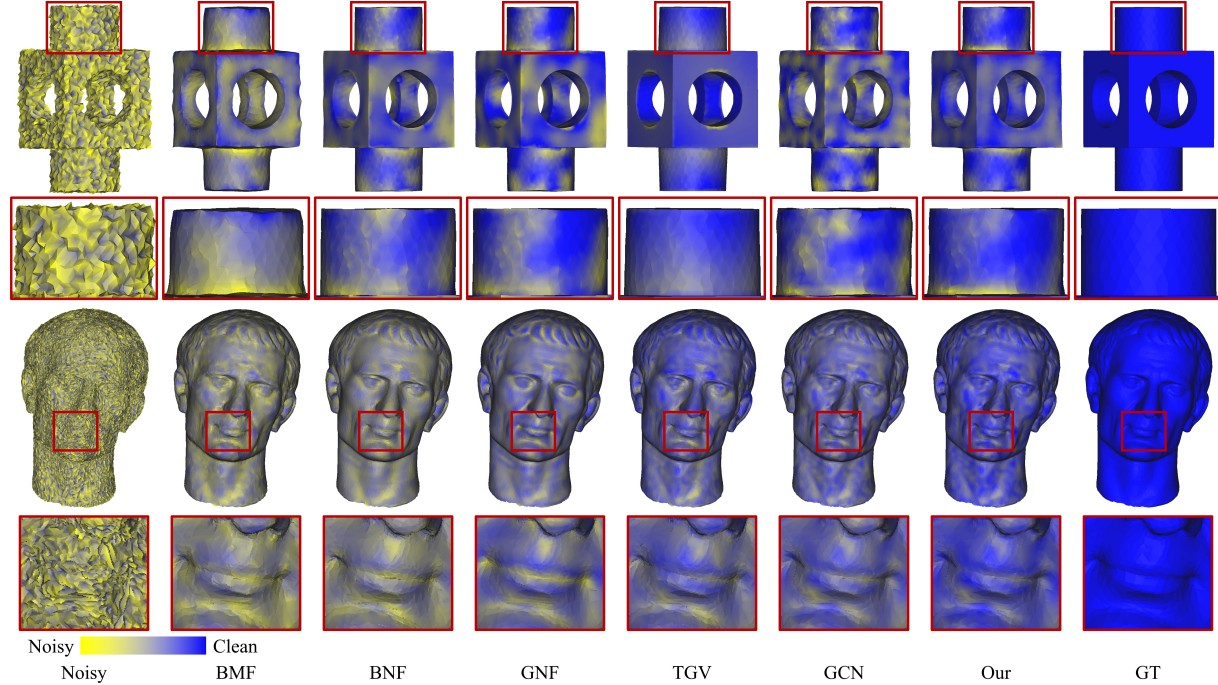

Noisy ▬▬▬ Clean

Noisy    BMF    BNF    GNF    TGV    GCN    Our    GT

**Figure 6: Representative denoised meshes on SynData. The distance from each denoised vertex to the ground truth mesh is color-coded as shown in the bar.**

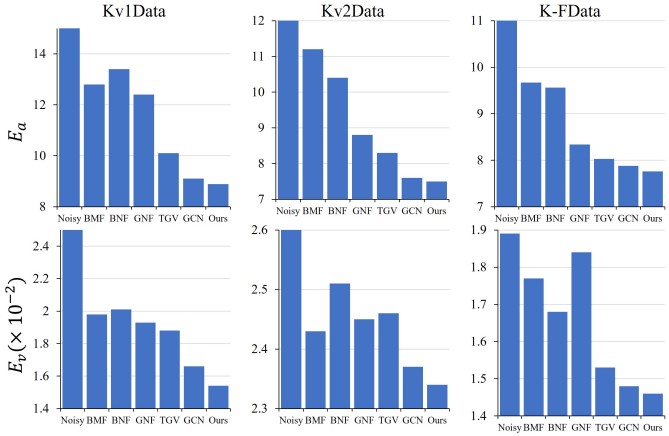

**Figure 7: The results of compared methods on Kinect series datasets.**

Here, $\mathbb{V}'$ and $\hat{\mathbb{V}}$ are the sets of denoised vertices and ground truth vertices. The smaller the $E_a$ and $E_v$, the better the performance.

## 5.2 Comparison Study

To evaluate the performance of FCL, we conduct qualitative and quantitative comparisons with state-of-the-art mesh denoising methods, including bilateral mesh filtering (BMF) [6], bilateral normal filtering (BNF) [37], guided normal filtering (GNF) [34],

**Table 2: The results of ablation experiments.**

| Variants | Simple meshes | | Complex meshes | |
|---|---|---|---|---|
| | $E_a$ | $E_v(\times 10^{-3})$ | $E_a$ | $E_v(\times 10^{-4})$ |
| w/o filtering | 5.15 | 2.52 | 6.21 | 9.13 |
| w/o $\mathbf{W}^N$ | 5.44 | 2.43 | 5.79 | 8.48 |
| w/o classifier | 4.74 | 2.31 | 4.91 | 8.32 |
| w/o multiply | 4.66 | 2.34 | 4.89 | 8.14 |
| Ours | **4.57** | **2.29** | **4.84** | **8.11** |

mesh total generalized variation (TGV) [15], and GCN-Denoiser (GCN) [23]. To ensure fair comparisons, we carefully select the best results obtained with fine-tuned parameters for BMF, BNF, GNF, and TGV as our competitors. For GCN, we train it using the same training data as FCL.

**Synthetic dataset**. In order to make the Hausdorff distances of different 3D meshes comparable, all meshes in SynData are scale-normalized through being divided by the diagonals of shape bounding boxes.

The experimental results are presented in Table 1. In this benchmark, our method achieves the smallest $E_a$ and $E_v$ values on complex meshes, indicating superior performance. For simple meshes, TGV achieves the best $E_a$ value, while our method performs better in terms of $E_v$. Overall, both TGV and our method demonstrate the best performance. Our method excels in handling complex meshes and remains competitive with TGV for simple meshes.

Figure 8: The denoising results of *boy* and *pyramid*, which are two representative meshes on the Kinect series datasets.

Two representative denoised meshes with color-coded errors are showcased in Fig. 6. In this test, the performances of BMF, BNF, and GNF are relatively similar, with GNF slightly outperforming the other two methods. TGV and GCN outperform these three methods, with TGV achieving impressive denoising results for the simple mesh in the top row. However, TGV tends to lose some texture features when processing complex meshes. GCN excels in handling complex meshes but struggles to recover sharp features. In contrast, FCL demonstrates the ability to recover sharp features in the top row and preserve fine-scale features in the bottom row, highlighting its efficacy.

**Real-scanned dataset**. The quantitative comparisons on the Kinect series datasets are presented in Fig. 7. It is evident that our method consistently outperforms the compared methods in all the datasets, including Kv1Data, Kv2Data, and K-FData. Two representative denoised meshes are displayed in Fig. 8, which suggests consistent conclusion with the results on synthetic data.

The results of the collected meshes are displayed in Fig. 9. It can be observed that all the compared methods effectively remove noise. However, our method performs better in preserving features, as demonstrated in the eye region of the *angel* mesh.

## 5.3 Ablation studies

To investigate the contribution of each component in FCL, we conduct four ablation experiments on the SynData dataset. The first variant, referred to as "w/o filtering", directly outputs the denoised normal instead of the filtering coefficients. This variant is implemented by adding two fully connected layers at the end of FCL. The purpose of this experiment is to verify the superiority of the filtering mechanism. The second variant, denoted as "w/o $\mathbf{W}^n$", excludes the first coefficient vector from the final coefficients. The purpose of this experiment is to investigate the importance of the noise-aware component in filtering coefficients. The third variant, referred to as "w/o classifier", eliminates the classifier, meaning that the category feature is not used in $\mathcal{M}_g$. The purpose of this experiment is to examine the importance of the category feature.

Table 3: The results of hyper-parameter selection experiments. $r_1$ denotes the number of rings for the input patch. $r_2$ is the number of rings for the patch used to label categories of faces. $I$ is the iteration number for generating $\mathcal{M}^M$. $n_c$ represents the category number. $\lambda$ is applied in $L$.

| Settings | $r_1$ | $r_2$ | $I$ | $n_c$ | $\lambda$ | Simple | | Complex | |
|---|---|---|---|---|---|---|---|---|---|
| | | | | | | $E_a$ | $E_v$ | $E_a$ | $E_v$ |
| Best | 3 | 2 | 20 | 4 | 0.001 | 4.57 | 2.29 | 4.84 | 8.11 |
| $V_1$ | **2** | 2 | 20 | 4 | 0.001 | 4.98 | 2.62 | 5.73 | 10.38 |
| $V_2$ | **4** | 2 | 20 | 4 | 0.001 | 4.61 | 2.16 | 5.13 | 9.34 |
| $V_3$ | 3 | **1** | 20 | 4 | 0.001 | 4.88 | 2.54 | 5.17 | 8.61 |
| $V_4$ | 3 | **3** | 20 | 4 | 0.001 | 4.63 | 2.48 | 4.96 | 8.33 |
| $V_5$ | 3 | 2 | **10** | 4 | 0.001 | 4.72 | 2.35 | 4.91 | 8.27 |
| $V_6$ | 3 | 2 | **30** | 4 | 0.001 | 4.64 | 2.38 | 4.94 | 8.39 |
| $V_7$ | 3 | 2 | 20 | **3** | 0.001 | 4.68 | 2.33 | 4.89 | 8.23 |
| $V_8$ | 3 | 2 | 20 | **5** | 0.001 | 4.61 | 2.36 | 4.90 | 8.26 |
| $V_9$ | 3 | 2 | 20 | 4 | **0.01** | 4.77 | 2.46 | 5.18 | 8.47 |
| $V_1 0$ | 3 | 2 | 20 | 4 | **0.0001** | 4.61 | 2.36 | 4.89 | 8.15 |

The last variant, represented as "w/o multiply", replaces the matrix multiply operation in $\mathcal{M}_g$ with a multi-layer perceptron. The results of these experiments are provided in Table 2. We can see that all of the variants performed worse than FCL, confirming the positive role played by each component in our method.

## 5.4 Studies on Hyper-parameters

In our proposed FCL, there are five hyper-parameters that need to be set: the size of the input patch (Subsection 3.1), the patch size for category labels (Subsection 3.3), the category number (Subsection 3.3), the iteration number for generating $\mathcal{M}^M$ (Subsection 3.2), and the $\lambda$ (Subsection 4.2) in the final loss function. These hyper-parameters are selected experimentally, and all experiments are conducted on the SynData dataset. Table 3 lists all of the experimented parameter settings. Each row represents a different

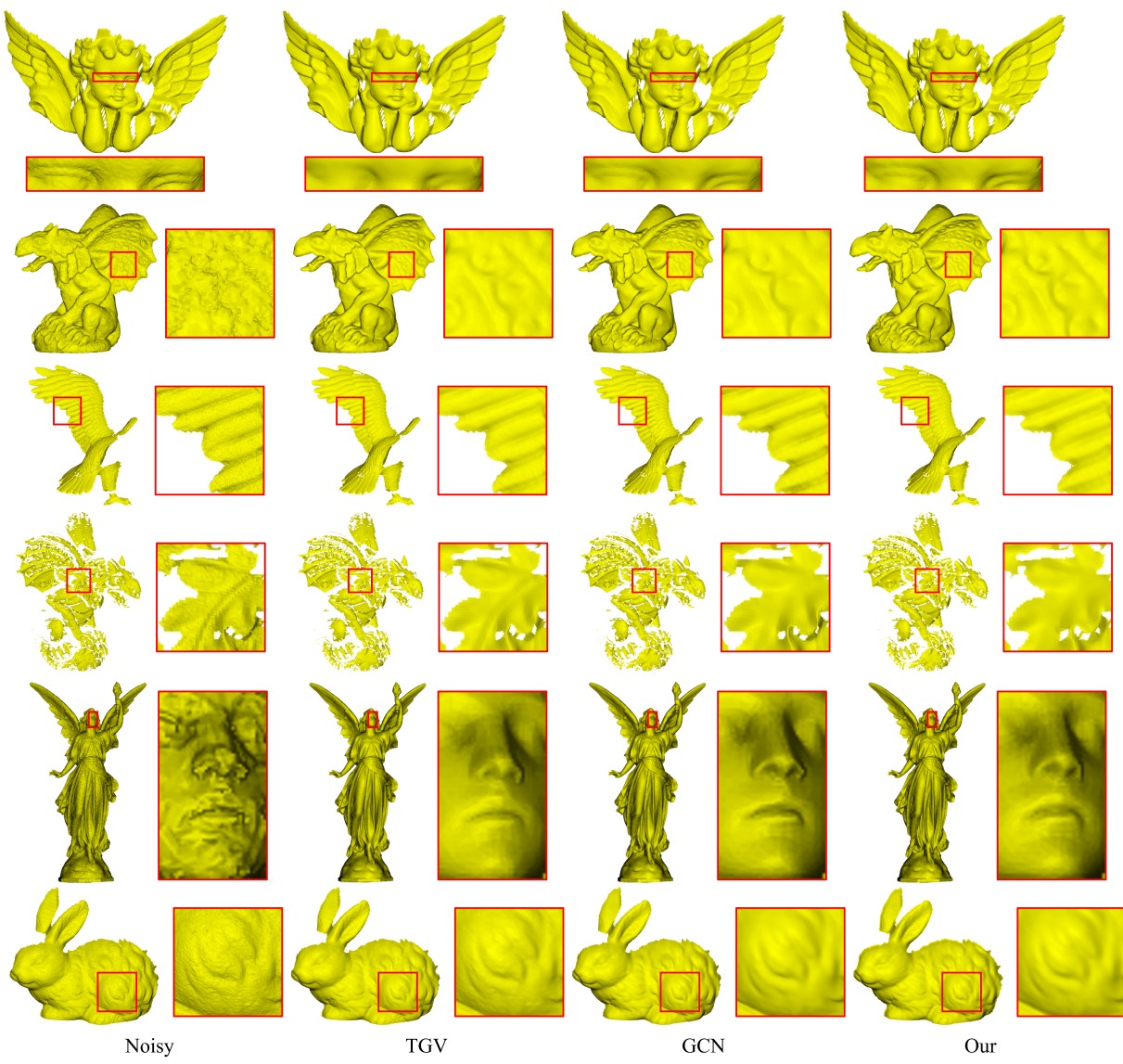

Noisy TGV GCN Our

**Figure 9: The denoising results of meshes collected from the internet.**

parameter setting, with the first row indicating the best configuration. To facilitate the selection of each parameter, each setting only changes one parameter, and the changed parameter in each setting is highlighted in bold. From the results, the size of input patches has a significant impact on the denoising performance. Overall, our experimental results demonstrate that FCL is an effective method for mesh denoising, and that careful parameter selection is important for achieving optimal denoising performance.

## 6 CONCLUSION

In this paper, we propose a novel Filtering Coefficient Learner (FCL for short) for mesh denoising by jointly considering noise intensity and geometric characteristics,. FCL produces filtering coefficients consisting of a noise-aware component and a geometry-aware component. The first component is inversely proportional to the noise intensity of each face, resulting in smaller coefficients for faces with stronger noise. The second component is derived based on two types of geometric features, where the category feature provides a global description of the input patch and the face-wise features complement the perception of local textures. Extensive experiments have validated the superior performance of FCL over state-of-the-art works in both noise removal and feature preservation. However, deep learning-based methods, including FCL, inherently encounter disadvantages when it comes to restoring sharp textures, in contrast to conventional methods. Additionally, the training of FCL is a meticulous and challenging process. These limitations warrant dedicated attention and efforts in the future to address.

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
