# OpenReview forum: "Mesh Denoising Using Filtering Coefficients Jointly Aware of Noise and Geometry"
_acmmm.org/ACMMM/2024/Conference — MM2024 Poster_

### Official Review · Reviewer_1kA4 · 2024-05-16

**Rating:** 4
**Confidence:** 1

**Summary:**

The author proposes a novel Filtering Coefficient Learner for mesh denoising by jointly considering noise intensity and geometric characteristics. Specifically, the proposed method includes (1) a noise intensity estimation module; (2) a geometry describing module; and (3) vertex updating. Experimental results demonstrate the advantages of FCL compared to some previous methods.

**Strengths:**

The writing is well organized, the experiments are comprehensive and show the effectiveness of the proposed method and its components. From the results, the method in this paper achieves state-of-the-art on both noise removal and feature preservation.

**Limitations:**

1. Some recent works [1,2] are missing in this paper.

     [1]. Zhang Y, Shen G, Wang Q, et al. Geobi-gnn: geometry-aware bi-domain mesh denoising via graph neural networks[J]. Computer-
     Aided Design, 2022, 144: 103154.

     [2]. Zhao Z, Tang W, Gong Y. Curvature‐driven Multi‐stream Network for Feature‐preserving Mesh Denoising[C]//Computer Graphics
     Forum. 2024, 43(1): e14993.

2. Is r in this paper the same definition as r1 in Table 3? If it is the same, please standardize the change. Otherwise, will this parameter have an important effect on the final performances, say change to other?
3. Please check the writing of the variables, such as i in equation (6).
4. The Category Probability plot in Figure 2 is too small and unclear.

**Suitability:**

2

---

### Official Review · Reviewer_8Wx3 · 2024-05-22

**Rating:** 4
**Confidence:** 2

**Summary:**

This paper proposes a normal denoising method, which regresses the weighting coefficients and utilizes the feature of the geometric classifier as guidance. Experimental results show better results than previous methods, especially on some complex objects.

**Strengths:**

1. This paper proposes to denoise the normal by estimating the coefficients.
2. A classifier is used to guide the feature extraction process.

**Limitations:**

1. Why regressing the coefficients is better than directly regressing the denoised normal? Though Fig. 3 shows the approximately linear relationship between the \alpha^M and \hat{\alpha}, I think this relationship holds only if the noise is sampled from some existing distributions, such as Gaussian noise in this paper. For real-world noise, I highly suspect that this relationship will be broken. Then, how to ensure the effectiveness of the coefficient regression.
2. There are four categories according to the maximum angle difference. How about the classification accuracy? Will the denoising performance be influenced by the classification accuracy? If so, the relationship between the denoiser and the classifier should be analyzed.
3. Is there any training and inference time analysis？
4. It seems the model is trained on a limited number of object meshes for a long time, i.e., 1,500 epochs, I think the model is easy to overfit on such a small-scale training set. Besides, the test set scale is also limited, e.g.,  5 simple objects and 5 complex objects. If the added noise is sampled from Gaussian distributions for the synthetic data, it is possible to generate more test cases by adding noise on various 3D object-level datasets.

**Suitability:**

2

---

### Official Review · Reviewer_9N5h · 2024-05-25

**Rating:** 3
**Confidence:** 3

**Summary:**

This paper propose a mesh denoising network which is jointly aware of noise and geometry. The authors design a novel filter named FCL, which generates coefficients to filter face normals. The authors conducted adequate experiments to demonstrate the validity of the method.

**Strengths:**

1.The idea of this paper is intriguing, proposing an effective FCL while jointly considering both noise features and geometric features.
2.The paper's approach to considering geometric features is quite novel. It utilize a method that extracts category features to differentiate between model surfaces with different maximum angle differences, rather than simply training on mixed datasets of different categories as in previous works。
3.The paper is logically clear and fluent in writing. The authors provide abundant experimental results and performance graphs.

**Limitations:**

1. In the introduction, the authors mention that the shortcomings of previous work lie in not explicitly considering both noise and geometry. However, this is inherently a denoising task. Why does the author believe that previous work did not consider the noise factor when completing denoising tasks? Furthermore, from the perspective of geometric features, GeoBi-GNN[1] has already proposed a Geometry-aware approach. The authors need to explain their motivation more clearly and reasonably.

2. The comparison algorithms in this paper are all from work prior to 2022, such as GCN, and even from 2003, such as BMF. Are there any more recent algorithms that could be used for comparison to demonstrate the effectiveness of the work?

3. I noticed that the network proposed by the author is mainly compared with GCN in the experimental section, and GCN is almost the best-performing method among the other comparative methods provided. However, the architecture of the network in this paper uses three GCN networks along with additional convolution, pooling, and fully connected layers, which means that the number of parameters is at least three times that of GCN. Yet, the improvement in performance does not seem to be that significant.

4. There are punctuation errors in the paper, such as the missing period at the end of the second paragraph in the Introduction. Additionally, the format of some of the figures is messy, with misalignment between rows and columns, such as in Figure 9.

5. In the comparative experiments section, the first two stages of training in this paper lasted for 500 epochs each, and the final stage lasted for 1000 epochs. However, the comparison algorithm GCN was trained for only 24 epochs in the first stage and 16 epochs in the second stage. Aligning the experimental settings would make the results more convincing.

[1]GeoBi-GNN: Geometry-aware Bi-domain Mesh Denoising via Graph Neural Networks.

**Suitability:**

3

---

### Official Review · Reviewer_Sg5B · 2024-05-27

**Rating:** 5
**Confidence:** 4

**Summary:**

The authors proposed a Filtering Coefficient Learner (FCL for short) for mesh denoising is developed, which delicately generates coefficients to filter face normals. Specifically, FCL produces filtering coefficients consisting of a noise-aware component and a geometry-aware component. The first component is inversely proportional to the noise intensity of each face, resulting in smaller coefficients for faces with stronger noise. For the effective assessment of the noise intensity, a noise intensity estimation module is designed, which predicts the angle between paired noisy-clean normals based on a mean filtering angle. The second component
is derived based on two types of geometric features, namely the category feature and face-wise features. The category feature provides a global description of the input patch, while the face-wise features complement the perception of local textures. Extensive experiments have validated the superior performance of FCL over state-of-the-art works in both noise removal and feature preservation.

**Strengths:**

- The authors proposed a Filtering Coefficient Learner (FCL for short) for mesh denoising is developed, which delicately generates coefficients to filter face normals.
- The noise-aware component and a geometry-aware component are interesting.

**Limitations:**

- The authors should report some false results obtained by the proposed method.
- How about the computational complexity of the proposed method when braining in two components, namely noise-aware component and a geometry-aware component.

**Suitability:**

2

---

### Meta-Review · Area_Chair_92Kk · 2024-07-03

**Recommendation:** Accept (Poster)
**Confidence:** 4

**Metareview:**

All reviews are generally positive towards this work. Acceptance is recommended.